# Minimally Invasive Intraoral Approach to Submandibular Lodge

**DOI:** 10.3390/jcm9092971

**Published:** 2020-09-14

**Authors:** Massimo Galli, Massimo Fusconi, Francesca Romana Federici, Francesca Candelori, Marco De Vincentiis, Antonella Polimeni, Luca Testarelli, Benedetta Cassese, Gabriele Miccoli, Antonio Greco

**Affiliations:** 1Department of Oral and Maxillofacial Science, Sapienza University of Rome, Via Caserta 6, 00161 Rome, Italy; massimo.galli@uniroma1.it (M.G.); f.federici@uniroma1.it (F.R.F.); antonella.polimeni@uniroma1.it (A.P.); luca.testarelli@uniroma1.it (L.T.); benedettacass@gmail.com (B.C.); 2Department of Sensory Organs, Division of Otorhinolaryngology, Sapienza University of Rome, Viale del Policlinico 161, 00185 Rome, Italy; massimo.fusconi@uniroma1.it (M.F.); francesca.candelori@uniroma1.it (F.C.); marco.devincentiis@uniroma1.it (M.D.V.); antonio.greco@uniroma1.it (A.G.)

**Keywords:** odontogenic abscess, cervicotomy, surgery, oral surgery, neck surgery, infection, phlegmon, oral cavity

## Abstract

The purpose of this study is to describe the Minimally Invasive Intraoral Approach (MIIA) performed on selected cases of abscesses and neck phlegmons of odontogenic origin when the infection has not spread beyond the inferior mandibular margin. This technique allows us to avoid cervicotomy by a direct approach to the abscess, draining it through the oral cavity. If the limits have already been crossed, then cervicotomy is necessary. The aim of the study is to show the surgical outcomes that we have achieved during a time span of two years, and to show the effectiveness of the MIIA and its results. We selected 66 patients with abscesses and neck phlegmons, from January 2018 to June 2020. Among these cases, five patients were excluded as it was not possible to recover medical records from database. The MIIA technique has been performed on 16 patients (26.2%) when a successful dental extraction and drainage of the submandibular lodge were accomplished. The patients who underwent the MIIA surgery have all perfectly healed and did not suffer from relapses during the follow-up. The results show the achievement of excellent healing, underlining the lower impact required by MIIA when compared to a more traditional approach through cervicotomy.

## 1. Introduction

Recently, an increasing number of patients with dental symptoms admitted to the Emergency Room (ER) has been recorded. These patients were diagnosed with abscesses and neck phlegmons of odontogenic origin, which required surgical treatment [1,2,3]. Unfortunately, in some cases the abscess had already spread to the point where it was not possible to solve the disease through simpler treatments, such as root canal therapy or periodontal approach, both aiming to resolve the infection and at the same time to save the involved teeth [4]. Most of the time, this kind of infection forces the patient to undergo complex surgery. Such an invasive surgical treatment is based on the extraction of an untreatable tooth and involves cervicotomy in order to clean the infected tissues and the placement of a Redon drainage. All of this inevitably results in a longer hospitalization. The procedure usually involves a submandibular incision that may lead to severities, such as the impairment of neuro-vascular structures, not to mention aesthetic issues relating to a healing neck wound [5,6]. The aim of this paper is to show an alternative surgical technique performed on selected cases. This technique avoids cervicotomy by means of a direct approach to the abscess, draining it directly through the oral cavity. Our hereby proposed Minimally Invasive Intraoral Approach (MIIA) can only be performed when the infection spreads up to the inferior mandibular margin and no further. As a matter of fact, the anatomical localization of the abscess and its spreading have to be pre-emptively evaluated with computerized tomography (CT) and the results then lead us to the most appropriate surgical technique. The goal of this study is to demonstrate the achievement of excellent healing after a MIIA, underlining the lower impact of MIIA when compared to the classic approach through cervicotomy. Moreover, we obtained our best results among all the patients selected for MIIA when positioning a drainage device to improve the anti-gravity discharge of liquid from the residual cavity by contraction of the suprahyoid muscles. This drainage consists of a latex glove finger, e.g. Penrose, placed between the alveolar bone and the mucoperiosteal flap. Specific antibiotic therapy remains absolutely essential for the complete cure of the infection. We proceed to show all of the surgical outcomes achieved during a time span of two years, alongside the criteria and algorithms used for surgical decision-making.

## 2. Experimental Section

This study has been approved by the Department of Oral and Maxillofacial Science IRB (Institutional review board), of the “Sapienza” University of Rome, and all participants signed an informed consent agreement. In this study, 66 patients admitted at the DAE (Department of Acceptance and Emergency) were considered from January 2018 to June 2020. These patients had been hospitalized to the Polyclinic of Rome “Umberto I” DAE, showing submandibular region swelling, where they were shortly after transferred to the Ear, Nose and Throat (ENT) Department and diagnosed with odontogenic abscess or neck phlegmon/abscess of odontogenic origin. Grounds for exclusion: five patients were excluded (admitted in 2019) as it was not possible to recover medical records from the database, while for another single patient [1] the infection involved a superior molar with consequent extraction and FESS(Functional Endoscopic Sinus Surgery). 61 patients have been tested in total, more precisely 25 cases admitted in 2018, 25 admitted in 2019 and 11 in 2020. 16 patients (24%) have undergone surgery for teeth extraction and trans-alveolar drainage without cutting the mucosa, since the CT scan results did not show any abscess located in the soft tissues. Among these, only one patient has undergone surgery with cervicotomy for an odontogenic abscess that occurred one week after the dental extraction. For 25 patients (40.9%) we performed teeth extraction and cervicotomy to drain pus, whereas in another three patients (4.9%) tracheostomy was considered necessary because the CT scan indicated that the infection had reached the space between the superior and deep cervical fascia (Figure 1).

The MIIA technique has been used on 16 patients (26.2%) in all, carrying out dental extraction and subsequent drainage of the submandibular lodge without a cervicotomy. Examining the CT scan results, the abscess–infected area had still not crossed the inferior medial side of the mandibula or the vertical plane passing through the posterior side of the lesser horn of the hyoid bone (Figure 2).

Each patient received assistance in reading and comprehension of the informed consent form, where the technique of MIIA was explained and where it was stated that, if necessary, cervicotomy would be performed. The MIIA can be executed when the abscess has reached, but not crossed, the inferior-medial side of the mandibula and/or the vertical plane, due also to the bone inflammation, spreading through the posterior side of the lesser horn of the hyoid bone. This last is a critical and hypothetical point where the hyoglossus muscle ends and the perivisceral space begins. It is important to notice that if the abscess is not lateral to the mylohyoid muscle, this implies that the above-mentioned limits have already been crossed and therefore a cervicotomy is necessary. Head and neck CT or MRI both with and without contrast are an essential support before the surgery, providing information regarding the area of the abscess. The surgery begins with an incision of the buccal and lingual mucosa of the infected teeth. It has been demonstrated that molars are generally more involved in odontogenic infections. The teeth most affected by pathological processes are molars. The first molars tend to erupt before the others. In addition, they are more likely to develop dental caries. The third molars are the last to erupt and they can easily experience dysodontiasis and pericoronitis. Moreover, in the molar region there is a higher index of dental plaque, due to the difficulties of maintaining good oral hygiene [7,8]. (Figure 3A). Continuing with the surgery, we proceed with the dissection of the mucogingival full-thickness flap (including mucosa, submucosa, muscles, and periosteum), which allows an entry to the contaminated submandibular lodge. This approach is safer and minimally invasive since it avoids injury to nerves and vessels that may occur medially in the lodge. The procedure continues with the extraction of the infected teeth, followed by the cleaning of the alveolar process by using surgical curettes. The lodge is carefully drained through suction and flushes. (Figure 3B). We then check if there is a pathway going from the socket to the abscess cavity using a blunted needle. In this exists, it will be cleaned with a sterile gauze bounded to the needle via silk suture. The gauze is prepared by soaking in a solution of hydrogen peroxide (50%) and saline (50%) and it is then brushed onto the sides both of the abscess cavity and the bone, passing through the newly originated pathway. Additional saline and peroxide flushes are then performed to mechanically reduce the bacterial load. Finally, we rinse the cavity generously, but only with saline to avoid complications from the use of peroxide. Eventually, a glove finger drainage is placed between the full-thickness flap, where is attached by sutures and by the bone, entering the abscess cavity. (Figure 3C). Thus, the cavity remains open and the gravitational drain is stimulated by the position taken by the patient lying on his side. Furthermore, the discharge is enhanced by the suprahyoid muscle-squeezing action during deglutition. The finger glove is removed after two–four days.

## 3. Results

The MIIA technique has been used on a total of 16 patients (26.2%), where successful dental extraction and drainage of the submandibular lodge was accomplished, all without a cervicotomy. Drainage was removed on the third post-operative day. On average, removed teeth were two per patient and the inferior are the most frequently extracted, especially molars 3.6, 3.7, 3.8, 4.6, 4.7, and 4.8. The patients who underwent MIIA surgery all perfectly healed and did not suffer from relapses during the follow-up. We have also evaluated and kept track of the drug therapy administered during hospitalization. Before admission, 34 persons were taking antibiotics, generally penicillin, without any association to specific gram-negative coverage and often without proper dosing and application. After admission 86% of the patients were treated with double antibiotic therapy in order to cover the broadest spectrum of aerobic and anaerobic bacteria, alongside with steroids to fight inflammation. The duration of treatment varied from five to ten days, and in only two cases was extended up to 21 and 25 days. The most frequently isolated germs in neck phlegmons and abscesses of odontogenic origin are predominantly Gram-positive organisms, such as staphylococci and streptococci, and lesser Gram-negative anaerobes deriving from periodontal bacterial flora [9] (Figure 4). Our protocol includes penicillin and cephalosporine or, in case of allergy, macrolides that have been shown to be effective against gram-positive organisms, always associated with a broad gram-negative coverage. We usually adopt the following scheme: amoxicillin 875 mg + clavulanic acid 125 mg or piperacillin 2 g + tazobactam 250 mg every 8 h, plus metronidazole 250 mg every 8 h or clindamycin 150 mg every 6 h. Ceftriaxone 1 g every 12 h could also be used instead of penicillin. Clarithromycin 500 mg every 12 h is used as an alternative to penicillin or cephalosporine when history of allergy is recorded. Betamethasone 1 mg per 10 kg is given to contain and reduce the inflammatory process and swelling. During hospitalization drugs are given intravenously followed by oral administration after discharge for at least seven days.

## 4. Discussion

The submandibular lodge (Figure 5) is a head and neck fascial space and derives from parotid fascia. In coronal section, it takes the shape of a triangular space showing three sides.

The superior-lateral side is formed by the fossa of the submandibular gland, located in the inner part of the jaw, immediately under the mylohyoid line. The superior-medial side is made up of mylohyoid muscle (anteriorly) and of hyoglossus muscle (posteriorly). Both muscles are covered by a smaller fascia derived from the suprahyoid fascia, also known as the superficial cervical fascia. In diverging on the top, these muscles form a virtual space that permits communication between the submandibular and the sublingual lodge. Finally, the inferior lateral side of our imaginary triangle is made up of cutis, subcutis, platysma muscle, and suprahyoid fascia. This fascia splits into two smaller fascia that, as mentioned above, form the superior-medial and inferior-lateral sides. These latter form the front end of the submandibular lodge at the point where they converge. The back end is made up of the intraglandular septum, also resulting from the same fascia that separates the submandibular lodge from that of the parotid [10,11,12]. Different interpretations exist regarding the anatomy of this space. We have already described the classic structure (Figure 6), but there are other variations. According to Charpy-Moresten (B), the inner layer of the superficial cervical fascia forms a pulley bypassing the digastric muscle’s tendon. The latter is then hooked onto the insertion of the stylohyoid muscle. According to Trolard-Decomps (C) alternatively, the deep cervical fascia rises above the hyoid bone, lining suprahyoid muscles and forming the inner side of the submandibular lodge. Finally, according to Truffert (D), the deep cervical fascia expands among the mylohyoid and stylohyoid muscles to form layers that cover the submandibular gland [13]. Certainly, anatomical variants can be found and the progress of the odontogenic abscess depends on these, eventually developing into lateral-cervical phlegmons. The anatomical variants B and D do not provide sufficient protection from the spreading of infection, making it necessary to perform cervicotomy a few hours after the occurrence of symptoms. There is no means of identifying the anatomical variant, either during surgery or before, via CT scan. Radiology only can define the area of the abscess. In this paper we want to show the effectiveness of the MIIA and its results, thoroughly explaining how the technique works, from the teeth’s extraction due to infection to the subsequent intraoral drainage of the submandibular lodge, all achieved with no need for cervicotomy.

This kind of surgery is possible only when the abscess, through bone inflammation, has reached (but not crossed) the triangular space of the jaw containing the mylohyoid muscle shown in Figure 1.

The MIIA can be executed when the abscess has reached—but not crossed—the inferior-medial side of the mandibula and/or the vertical plane, also due to bone inflammation spreading through the posterior side of the lesser horn of the hyoid bone. This last is a critical point, i.e. where the hyoglossus muscle ends and the perivisceral space begins. It is important to notice that if the abscess is not lateral to the mylohyoid muscle, this implies that the above-mentioned limits have already been crossed and therefore a cervicotomy is necessary. Head and neck CT or MRI both with and without contrast are an essential support before surgery, providing information regarding the area of the abscess.

This virtual space ensures that the infection does not overstep the anatomical limits towards the adjacent visceral tissues, which would then lead to the need for cervicotomy. Such limits are enclosed by the mylohyoid muscle—defining the external side of the sublingual region and the tongue base—by the inferior side of the jaw, and by the lesser horn of the hyoid bone. The latter corresponds to the posterior side of the submandibular lodge which is open at its rear to allow the passage of sublingual vessels (artery and vein), the lingual nerve, and the hypoglossal nerve [14,15].

Precisely, this rear opening is the most critical point—among the other anatomical structures mentioned earlier—since it is often too weak to contain the infection that will therefore easily spread under the effect of gravity. Obviously if the CT scan shows an infection that has crossed these limits, the MIIA would be an inadequate measure to prevent a further spreading of the infection and it would be necessary to proceed with a cervicotomy.

We have already described the anatomical variants of the submandibular lodge. When the infection fills the virtual spaces of the superficial and deep cervical fascia (B,D) it could potentially spread into the visceral region. On the contrary, when the case of an empty submandibular lodge occurs, if we operate in time we are able to drain the deep sides of the lodge, eluding the risk of the infection slipping through an intraoral pathway, which could potentially turn from an odontogenic abscess into a neck’s phlegmon, for which a cervicotomy would be required [10].

## 5. Conclusions

The MIIA, in selected cases, can lower the impact of the surgery, consequently reducing the length of hospitalization and cutting health costs. When abscess occurs and does not cross the previously described anatomical limits, we suggest the use of this technique to obtain a shorter post-operative recovery.

## Figures and Tables

**Figure 1 jcm-09-02971-f001:**
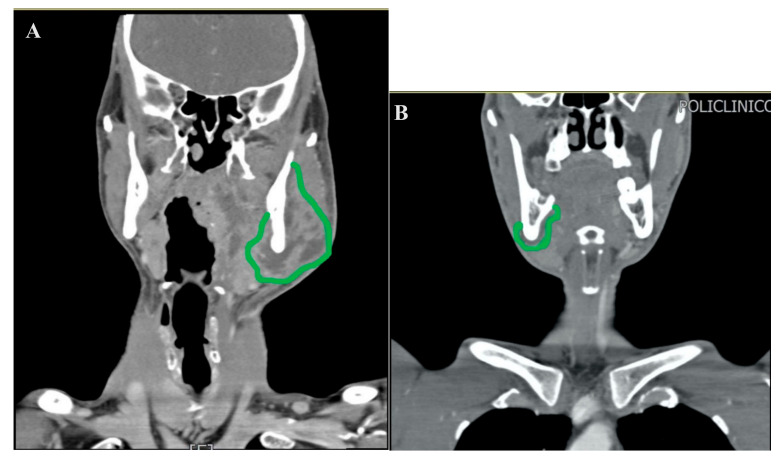
CT scan of coronal sections (**A**,**B**). The green line delimits the abscess’s area, showing the crossing of the inferior-medial side of the mandibula.

**Figure 2 jcm-09-02971-f002:**
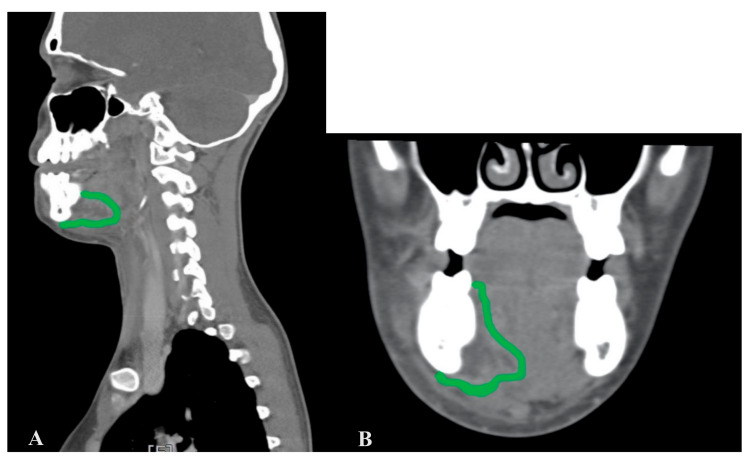
CT scan of sagittal (**A**) and coronal (**B)** sections. The green line delimits the abscess’s area that does not cross the inferior-medial side of the mandibula.

**Figure 3 jcm-09-02971-f003:**
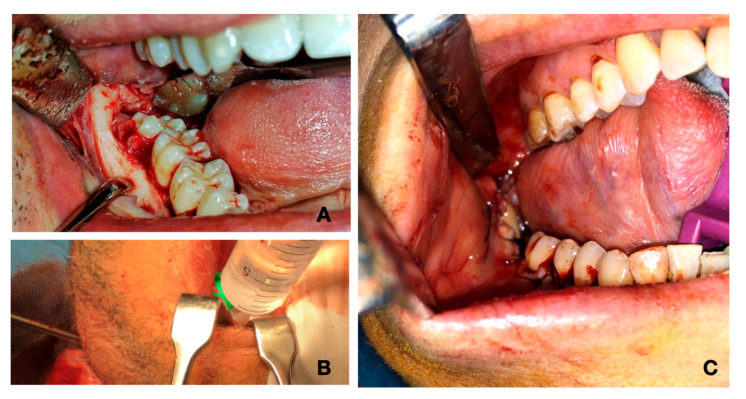
(**A**) Incision of buccal and lingual mucosa of the infected teeth. (**B**) Abscess cavity cleaning. (**C**) Glove finger drainage placed between the flap and the bone.

**Figure 4 jcm-09-02971-f004:**
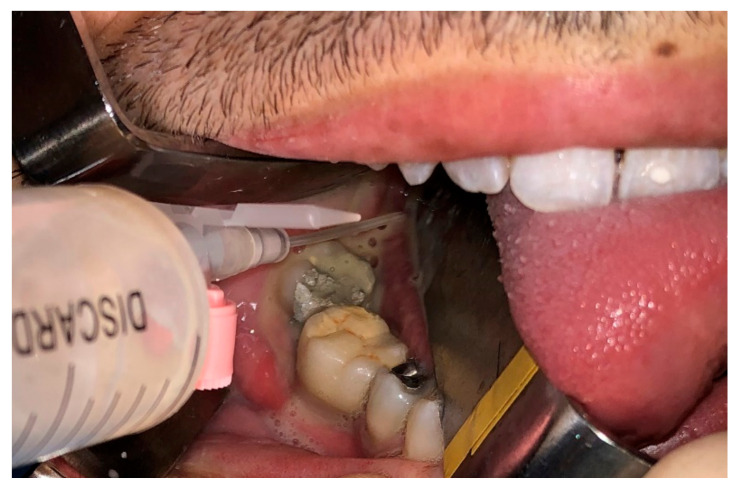
Sample of purulent material from the abscess.

**Figure 5 jcm-09-02971-f005:**
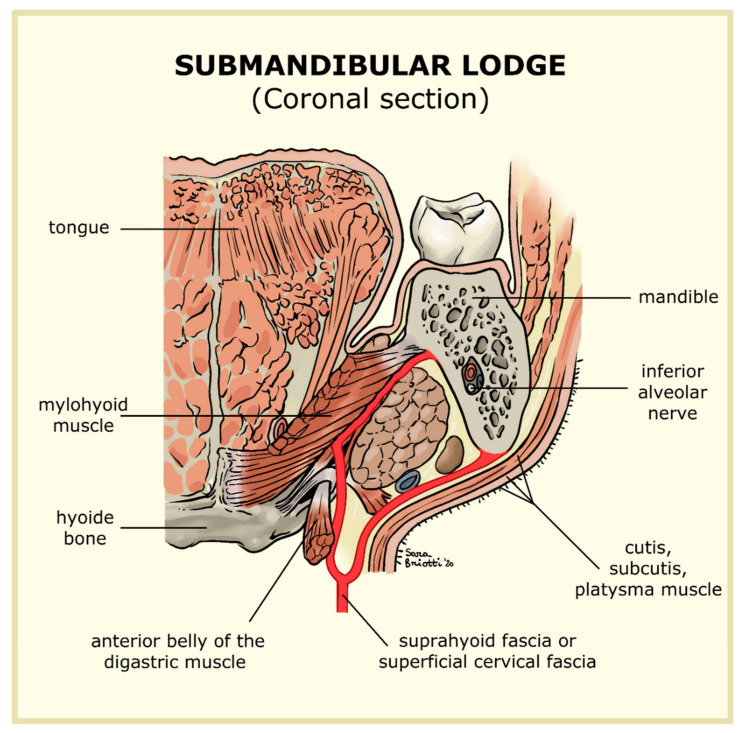
Coronal section of the submandibular lodge. Highlighted in red is the Suprahyoid Fascia.

**Figure 6 jcm-09-02971-f006:**
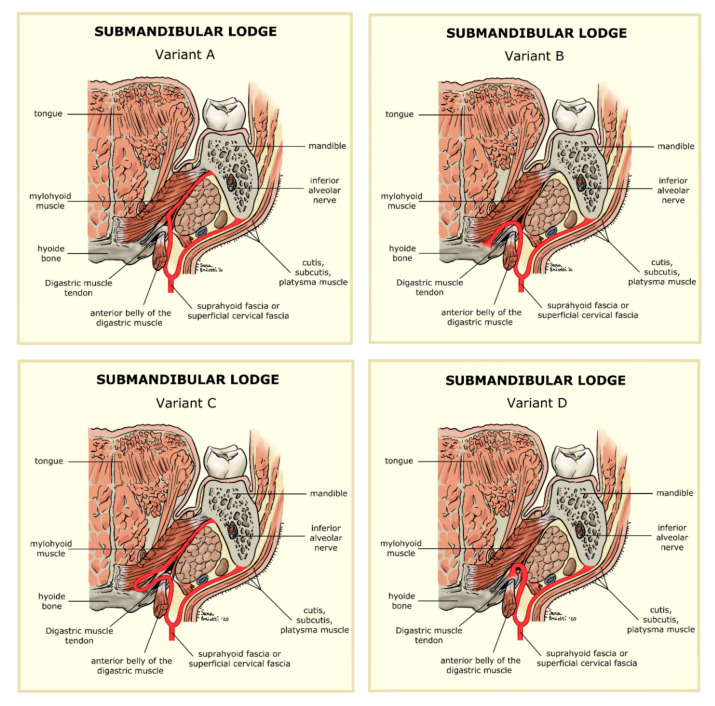
Anatomical variants of the Suprahyoid Fascia. (**A**) regular anatomy, (**B**) Charpy-Moresten anatomical variant, (**C**) Trolard-Decomps anatomical variant, (**D**) Truffert anatomical variant.

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
