# Peer review of "Minimally Invasive Intraoral Approach to Submandibular Lodge"

_jcm, 2020, doi:10.3390/jcm9092971_

Round 1

Reviewer 1 Report

A minimal invasive and esthetic-friendly surgical treatments are the trending in dentistry nowadays. This study aimed at MIA to replace traditional cervicotomy in some cases. The topic has significant clinical value; however, the study design and the manuscript need major revisions.

  1. Case selection is extremely important for the clinician to choose between MIA and cervicotomy. Clinical case selection criteria for MIA including radiographic guideline?
  2. Line 118-121, please demonstrate these descriptions in anatomy fissures including variations ABCD.
  3. Line 125-143. Please make it into a procedure protocol with demonstrations of figures. Also, procedure description belongs to methodology instead of discussion.
  4. List representative clinical photos to demonstrate MIA in order to thoroughly explain how the technique works
  5. 12 cases are small numbers to conclude if a new technique is reliable. Please enlarge the study sample size.
  6. Line 147-155. Please re-evaluate your results. The study aims to discuss the result and procedures of MIA. Only 12 patients undergone MIA treatment. Please exclude non-MIA patients from your results.
  7. Line 150-152. Please provide details of antibiotic and steroid treatment after MIA because it is a necessary part of the treatment protocol, including medicine name, dose, frequency.
  8. Please specify the antibiotic guideline after MIA.

Author Response

Dear  reviewer, 

thank you for your comments concerning our manuscript, they are very helpful to improving our paper. We have analyzed the comments and made the corrections you have requested. You can find the adjustments in the attachment: you'll find written in green the additions required such as drugs administration and references, in red the changes we made to the content already written. we also have added photos and corrected typo errors. 

Is it possible to change the order of the name? in the attachment I've already switched the name (Francesca Candelori) in fourth place instead of the ninth. 

Sincerely,

Dr. Miccoli

In response to point five:

5. Now a day an increasing number of patients with abscesses or neck's phlegmons is been recording in the italian's ER. Anyway only a few decine are admitted in the our department every year. The study considered admitted patients between January 2018 and January 2020. We have extended the analysis to June 2020, adding 9 cases. Among these only 4 have undergone MIIA. 

Reviewer 2 Report

I read with great interest the proposal made by the authors on a new minimally invasive surgical approach to the submandibular lodge.
This article is well written, however some elements could be improved for reading comfort

Major revisions:
Indeed, the structure of the article is flawed. The introduction is well written, although I wonder if an anatomical description of the lodge would be better in the introduction (e.g. Figure 1).
The methodology part is insufficient.I will put the elements given by the authors in discussion and add details on the surgical approach and clinical photographs to illustrate for the readers.
The results part is insufficiently described because elements of results are also discussed (drugs, follow-up information ...).
Finally, the discussion part will have to be rewritten in part according to the modifications mentioned above.

Minor revisions:
- MIA and MIIA cohabit in the text, only one of these terms should exist.
- L90, typo error, a brace is missing.
- Regarding the use of oxygen peroxide, the authors had no complications?

In the end this article is very interesting but more details should be given on the surgical technique.

Author Response

Dear reviewer, 

thank you for your comments. We agree with your suggestions and we have made the corrections you have requested. We have changed the structure, correct typo errors, improved methodology and results, and added photos. You can find all the adjustments in the attachment. We left the description of the submandibular lodge in the discussion to comparate with the anatomical variants. 

Regarding the hydrogen peroxide use, we never came across any kind of trouble because we are used to dilute it with saline (50%) and rinse generously again the cavity only with saline. 

You can find the adjustments in the attachment: you'll find written in green the additions required such as drugs administration and references, in red the changes we made to the content already written. 

Is it possible to change the order of the name? in the attachment I've already switched the name (Francesca Candelori) in fourth place instead of the ninth. 

Sincerely,

Dr. Miccoli

Round 2

Reviewer 1 Report

The manuscript has been significantly improved after the major revision. The methodology of clinical management of the MIIA was clearly described and demonstrated with figures of one case. Discussion focused on the limitations of MIIA and advantages. Overall, this study provided valuable clinical statical data as a clinical guidance.